# Sleep Quality Disturbances Are Associated with White Matter Alterations in Veterans with Post-Traumatic Stress Disorder and Mild Traumatic Brain Injury

**DOI:** 10.3390/jcm12052079

**Published:** 2023-03-06

**Authors:** Philine Rojczyk, Johanna Seitz-Holland, Elisabeth Kaufmann, Valerie J. Sydnor, Cara L. Kim, Lisa F. Umminger, Tim L. T. Wiegand, Jeffrey P. Guenette, Fan Zhang, Yogesh Rathi, Sylvain Bouix, Ofer Pasternak, Catherine B. Fortier, David Salat, Sidney R. Hinds, Florian Heinen, Lauren J. O’Donnell, William P. Milberg, Regina E. McGlinchey, Martha E. Shenton, Inga K. Koerte

**Affiliations:** 1Psychiatry Neuroimaging Laboratory, Department of Psychiatry, Brigham and Women’s Hospital, Harvard Medical School, Boston, MA 02145, USA; 2cBRAIN, Department of Child and Adolescent Psychiatry, Psychosomatics, and Psychotherapy, Ludwig-Maximilians-University, 80336 Munich, Germany; 3Department of Psychiatry, Massachusetts General Hospital, Harvard Medical School, Boston, MA 02114, USA; 4Department of Neurology, Ludwig-Maximilians-University, 81377 Munich, Germany; 5Department of Radiology, Brigham and Women’s Hospital, Harvard Medical School, Boston, MA 02115, USA; 6Department of Software Engineering and IT, École de Technologie Supérieure, Montreal, QC H3C 1K3, Canada; 7Translational Research Center for TBI and Stress Disorders (TRACTS) and Geriatric Research, Education and Clinical Center (GRECC), VA Boston Healthcare System, Boston, MA 02130, USA; 8Department of Psychiatry, Harvard Medical School, Boston, MA 02215, USA; 9Neuroimaging Research for Veterans (NeRVe) Center, VA Boston Healthcare System, Boston, 02115 MA, USA; 10Athinoula A. Martinos Center for Biomedical Imaging, Massachusetts General Hospital, Department of Radiology, Boston, MA 02129, USA; 11Department of Neurology, Uniformed Services University, Bethesda, MD 20814, USA; 12Department of Pediatric Neurology and Developmental Medicine and LMU Center for Children with Medical Complexity, Dr. von Hauner Children’s Hospital, Ludwig-Maximilians-University, 80337 Munich, Germany; 13Graduate School of Systemic Neurosciences, Ludwig-Maximilians-University, 82152 Munich, Germany

**Keywords:** sleep disturbances, mTBI, PTSD, dMRI, PSQI, military

## Abstract

Sleep disturbances are strongly associated with mild traumatic brain injury (mTBI) and post-traumatic stress disorder (PTSD). PTSD and mTBI have been linked to alterations in white matter (WM) microstructure, but whether poor sleep quality has a compounding effect on WM remains largely unknown. We evaluated sleep and diffusion magnetic resonance imaging (dMRI) data from 180 male post-9/11 veterans diagnosed with (1) PTSD (*n* = 38), (2) mTBI (*n* = 25), (3) comorbid PTSD+mTBI (*n* = 94), and (4) a control group with neither PTSD nor mTBI (*n* = 23). We compared sleep quality (Pittsburgh Sleep Quality Index, PSQI) between groups using ANCOVAs and calculated regression and mediation models to assess associations between PTSD, mTBI, sleep quality, and WM. Veterans with PTSD and comorbid PTSD+mTBI reported poorer sleep quality than those with mTBI or no history of PTSD or mTBI (*p* = 0.012 to <0.001). Poor sleep quality was associated with abnormal WM microstructure in veterans with comorbid PTSD+mTBI (*p* < 0.001). Most importantly, poor sleep quality fully mediated the association between greater PTSD symptom severity and impaired WM microstructure (*p* < 0.001). Our findings highlight the significant impact of sleep disturbances on brain health in veterans with PTSD+mTBI, calling for sleep-targeted interventions.

## 1. Introduction

Approximately 23% of military service members returning from deployment to Iraq and Afghanistan are subsequently diagnosed with post-traumatic stress disorder (PTSD) [1], making it one of the most common psychiatric diagnoses in veterans [2]. Additionally, 12–35% of service members sustain a mild traumatic brain injury (mTBI) [3,4], which increases the risk of developing or exacerbating PTSD symptom severity [5,6,7]. While poor sleep quality is highly prevalent in veterans in general [8], it is particularly important in the context of PTSD and mTBI. In fact, poor sleep quality is a hallmark symptom of PTSD [9,10], highly prevalent after mTBI [11,12,13], and has been associated with increased symptom severity [14] and slower recovery from both PTSD and mTBI [13]. Alarmingly, sleep quality disturbances may remain present for years after a traumatic experience [15], adversely impact quality of life [14,16], and respond poorly to treatment-as-usual [17,18]. However, the pathomechanisms underlying the compounding effects of poor sleep quality in those with PTSD and mTBI are not fully understood, impeding the development of proper diagnostic and treatment protocols.

Evidence suggests that sleep is directly linked to brain homeostasis and is essential to preserve the environment the brain requires to function optimally [19]. Interestingly, sleep seems particularly relevant for white matter health, with animal studies showing that the number of myelin-forming proliferating oligodendrocytes doubles during sleep [20]. Furthermore, sleep impairments have been related to brain volume loss [21,22], reduced neurogenesis, and reduced cortical activation [23,24].

Magnetic resonance imaging (MRI) allows for the in vivo, three-dimensional investigation of brain structure, and thus, provides a vital avenue to study the pathomechanisms underlying poor sleep quality, PTSD, and mTBI in military veterans [25]. Diffusion-weighted MRI (dMRI) studies revealed white matter (WM) microstructure alterations associated with PTSD [26,27,28,29,30,31,32,33,34,35,36,37,38,39,40,41,42,43], mTBI [44,45,46,47,48,49,50], and comorbid PTSD+mTBI [51,52,53,54]. Moreover, various studies showed widespread alterations of white matter microstructure in association with poor sleep quality in otherwise healthy individuals. Most studies utilized fractional anisotropy (FA) as a marker of white matter microstructural integrity and tissue organization and found lower FA related to poorer sleep quality in frontal, temporal, parietal, and occipital regions [55,56,57,58,59,60,61,62,63,64,65]. A few studies have also linked white matter alterations to impaired sleep quality in individuals with PTSD [66] and mTBI [67,68], revealing associations between poor sleep quality and decreased FA in several main white matter fiber tracts. However, the link between sleep quality and WM microstructure in the context of PTSD and mTBI remains largely unknown.

The current study addresses the understudied impact of sleep quality disturbances on brain structure in the context of PTSD and mTBI. We assess the relationship between sleep quality, PTSD, mTBI, and WM microstructure, leveraging a large sample of veterans returning from deployment to Iraq and Afghanistan (*N* = 180). As highlighted above, earlier studies have consistently demonstrated the impact of PTSD, mTBI, or comorbid PTSD+mTBI on brain structure and function. Therefore, we followed a previous publication [69] and examined sleep quality in groups of veterans with PTSD, mTBI, comorbid PTSD+mTBI, or no history of PTSD or mTBI. Next, we assessed the associations between PTSD, mTBI, sleep quality, and WM microstructure in the combined sample and the four groups. Our central hypotheses were that veterans with PTSD, mTBI, or comorbid PTSD+mTBI would experience poorer sleep quality than veterans without a history of PTSD or mTBI. We further hypothesized that PTSD, mTBI, and poorer sleep quality were related to WM microstructure. We subsequently conducted post-hoc mediation analyses based on our primary analyses to examine the interactions between PTSD, mTBI, sleep quality, and WM microstructure.

## 2. Materials and Methods

### 2.1. Participants

The Translational Research Center for TBI and Stress Disorders (TRACTS) study is a longitudinal prospective cohort study of Operation Enduring Freedom (OEF)/Operation Iraqi Freedom (OIF) military service members [70].

Out of the first 384 consecutively recruited service members at the VA Boston Healthcare System Jamaica Plain Campus, 278 participants underwent an MRI assessment and consented to share their data with investigators outside of TRACTS. Twenty-five cases did not pass the visual neuroimaging data quality control due to excessive motion or scanner artifacts. Additional reasons for the exclusion of participants can be found in Appendix A. The final sample for the current project consisted of 180 participants, which were classified into four groups based on lifetime diagnoses of PTSD and mTBI: PTSD only (*n* = 38), mTBI only (*n* = 25), comorbid PTSD+mTBI (*n* = 94), and no history of PTSD or mTBI (*n* = 23). We opted for this grouping approach based on previous studies in veteran populations [52,71]. We used cumulative lifetime diagnoses, as previous studies demonstrate a stronger influence on GM and WM structure when considering lifetime diagnoses [51,72], suggesting that disorder-related neurobiological effects persist or even increase over time.

### 2.2. Diagnostic and Clinical Assessment

#### 2.2.1. Assessment of PTSD

Lifetime PTSD diagnosis and current symptom severity were assessed according to the 30-item Clinician-Administered PTSD Scale for DSM-IV (CAPS-IV) [73]. To assess PTSD symptom severity separate from sleep quality, we removed two items (i.e., difficulty sleeping and recurrent distressing dreams) from the scale and calculated a sleep-corrected PTSD total score, in line with prior work [74].

#### 2.2.2. Assessment of mTBI

The Boston Assessment of TBI-Lifetime (BAT-L) [75] was administered to diagnose mTBI and to rate the cumulative lifetime mTBI burden. The BAT-L distinguishes mild, moderate, and severe TBIs, where a mild TBI is classified as loss of consciousness not exceeding 30 min and where posttraumatic amnesia or an altered mental status must not exceed 24 h. The BAT-L classifies mild TBI into stages 1–3, where a higher stage refers to greater mTBI severity. Using this assessment tool, a total mTBI burden score was computed from the number and severity of all mTBIs pre-, during, and post-deployment. Pre-deployment mTBIs included mTBIs before enlistment. MTBIs during deployment referred to all deployments if deployed multiple times.

#### 2.2.3. Assessment of Sleep Quality

Current sleep quality was assessed using the Pittsburgh Sleep Quality Index (PSQI) [76], an 18-item self-report questionnaire measuring subjective sleep quality, sleep latency, sleep duration, habitual sleep efficiency, sleep disturbances, use of sleep medication, and daytime dysfunction on seven subscales. Here, we follow a previously reported and validated approach suggesting that the seven subscales of the PSQI are best represented by three factors: sleep efficiency, perceived sleep quality, and daily disturbances [77] (Appendix A).

#### 2.2.4. Assessment of Comorbid Psychiatric Disorders

The non-patient research version of the Structured Clinical Interview for DSM-IV Axis I Disorders (SCID-I/NP) [78] was used to diagnose comorbid lifetime psychiatric disorders.

#### 2.2.5. Assessment of Warzone-Related Stress

Warzone-related stress was assessed using the Deployment Risk & Resilience Inventory-2 (DRRI-2) [79].

### 2.3. Magnetic Resonance Imaging

#### 2.3.1. Image Acquisition

DMRI data were acquired on a 3-Tesla Siemens TIM Trio scanner (Siemens Healthineers, Erlangen, Germany) at the VA Medical Center in Boston using a single-shot echo-planar sequence with a twice-refocused spin-echo pulse. The following sequence parameters were applied: 64 axial slices with no inter-slice gap, 60 gradient directions with a b-value of 700 s/mm^2^, and 10 additional scans with b = 0 gradients, TR = 10,000 ms, TE = 103 ms, voxel size = 2 × 2 × 2 mm^3^, and FOV = 256 mm^2^.

#### 2.3.2. Image Pre-Processing

The dMRI data were processed in several steps using an in-house image processing pipeline (https://github.com/pnlbwh/pnlutil/blob/master/pipeline/README.md, accessed on 1 October 2018). First, the images were axis-aligned, centered, motion-, and eddy current-corrected utilizing the FMRIB Software Library (version 5.1, https://fsl.fmrib.ox.ac.uk/fsl/fslwiki/, accessed on 1 October 2018) [80,81]. Image quality was checked for artifacts using 3D Slicer (version 4.5, http://www.slicer.org, accessed on 1 October 2018) [82], leading to the exclusion of 25 participants (due to severe motion artifact or signal dropout). DMRI brain masks were created using SlicerDMRI [83,84] and corrected manually where necessary.

#### 2.3.3. WM Fiber Clustering

We conducted WM fiber clustering utilizing an open-source pipeline, *whitematteranalysis software* (https://github.com/SlicerDMRI/whitematteranalysis, accessed on 1 May 2019), to perform fiber tract parcellation and extraction automatically. The white matter fiber clustering method extracts fiber tracts from the entire brain by grouping tracts based on their anatomical shape and spatial location. This method is significantly improved compared to previous automated fiber tracking methods, which can only extract the main fiber tracts, failing to cover the entire brain’s white matter (i.e., including the cerebellum and superficial tracts). In addition, it was successfully applied in several recent studies [85,86,87,88,89,90,91,92,93,94], demonstrating high test-retest reproducibility [95] and robustness to anatomical variability [96].

First, a two-tensor whole-brain unscented Kalman Filter (UKF) tractography was computed (https://github.com/pnlbwh/ukftractography, accessed on 1 May 2019) [97,98]. A two-tensor model was chosen to account for crossing fibers [99,100]. The first tensor is associated with the main direction of a fiber tract, while the second tensor represents crossing fibers. We performed qualitative and quantitative quality checks of the generated tractography data for all subjects using the *whitematteranalysis software* quality control tool (https://github.com/SlicerDMRI/whitematteranalysis, accessed on 1 May 2019). Previous studies demonstrated that the UKF method is highly consistent [101] and more sensitive than single-tensor tractography [102,103,104].

Next, we identified white matter fiber tracts for each subject using the White Matter Analysis (WMA) package for tract parcellation. WMA is based on a neuroanatomist-curated white matter atlas (http://dmri.slicer.org/atlases/, accessed on 1 May 2019) [101] and applies machine learning to identify fiber tracts in an individual [101,105,106]. This approach enabled us to substantially reduce the known tractography issue of false-positive tracking, increasing the repeatability of white matter parcellation [107]. False positive fibers in the atlas have been annotated and marked as to be excluded based on expert neuroanatomical judgment [101]. For each subject, atlas-based white matter parcellation [96,101,106] was performed, registering the tractography to the atlas space. The similarity between the fibers in the atlas and the fibers of an individual was quantified, used to classify the fibers into a cluster, and finally assigned to the corresponding tract in the atlas.

As highlighted in the introduction, we expected a widespread effect of sleep on WM microstructure [55,56,57,58,63]. For our primary analyses, we, therefore, opted to merge the entire brain’s fiber tracts into one whole-brain WM variable by appending all WM tracts into one large tract. For supplementary analyses, the main white matter fiber tracts (left/right arcuate fasciculus, cingulum bundle, inferior longitudinal fasciculus, inferior occipito-frontal fasciculus, superior longitudinal fasciculus, uncinate fasciculus, and corpus callosum) were extracted. To ensure there were no individual participants with outlier values, we performed a quantitative quality assessment of the number of fiber streamlines. Moreover, each participant’s whole-brain white matter tracts were visually evaluated, following standardized guidelines [85,89,108,109,110,111]. All data successfully passed the quality check.

#### 2.3.4. Diffusion Parameter Extraction

We used free-water (FW) imaging to obtain a voxel-wise whole-brain free-water corrected fractional anisotropy (FA_T_) value for each individual. By separating the MRI signal into two compartments [112], FW imaging is able to eliminate partial volume with extracellular FW (e.g., caused by CSF contamination, edema, or atrophy) in each voxel. Given the correction for FW, FA_T_ serves as a more accurate marker for cellular WM structure than the conventional FA measure [113].

### 2.4. Statistical Analysis

Statistical analyses were performed using IBM SPSS Statistics 27. We created figures using R 4.0.3, GraphPad Prism 9, Python 3.10.2, and PowerPoint. We applied a hierarchical statistical approach, conducting all analyses in the total sample and if significant in the four groups (PTSD, mTBI, comorbid PTSD+mTBI, no history of PTSD or mTBI; Figure 1). All analyses included age as a covariate and were corrected for multiple comparisons, as detailed in Figure 1.

#### 2.4.1. Group Differences in Sleep Quality

First, we wanted to examine if sleep quality differed according to veterans‘ histories of PTSD and mTBI. We conducted one ANCOVA, including group (PTSD, mTBI, comorbid PTSD+mTBI, no history of PTSD or mTBI) as the independent variable and global sleep quality as the dependent variable. If the overall ANCOVA was significant (*p* < 0.05), we performed post-hoc comparisons for the four groups. If the group comparisons for global sleep quality were significant (*p* < 0.05/4), we post-hoc compared sleep efficiency, perceived sleep quality, and daily disturbances between the four groups.

#### 2.4.2. PTSD, mTBI, and Sleep Quality

Next, we calculated one regression analysis in the total sample, including PTSD symptom severity and mTBI burden as the independent variables and global sleep quality as the dependent variable. In the case of significant associations between PTSD symptom severity, mTBI burden, and global sleep quality (*p* < 0.05), we repeated the regression analyses within the four groups separately. If the regression model was found significant in one of the groups (*p* < 0.05/4), we performed three additional regression analyses, including PTSD symptom severity/mTBI burden as the independent variable and sleep efficiency, perceived sleep quality, and daily disturbances as dependent variables respectively.

#### 2.4.3. Sleep Quality and WM Microstructure

Next, we conducted a regression analysis in the total sample, including global sleep quality as the independent variable and whole-brain FA_T_ as the dependent variable. In the case of a significant association (*p* < 0.05), we repeated the regression analysis within each group. If found significant in one of the groups (*p* < 0.05/4), we performed three additional regression analyses, including sleep efficiency, perceived sleep quality, and daily disturbances as the independent variables and whole-brain FA_T_ as the dependent variable.

In addition, we computed supplementary analyses with the FA_T_ of the main WM fiber tracts as the dependent variables (Appendix A).

#### 2.4.4. Sleep Quality as a Mediator between PTSD Symptom Severity and WM Microstructure

Given the significant associations between PTSD symptom severity and perceived sleep quality and between perceived sleep quality and whole-brain FA_T_, we performed a post-hoc mediation analysis. Here, we assessed whether perceived sleep quality (mediator) mediated the association between PTSD symptom severity (independent variable) and whole-brain FA_T_ (dependent variable). The model was calculated using the Hayes PROCESS macro [114] for SPSS (model 4), which follows a nonparametric bootstrapping procedure based on *n* = 5000 samples and a 95% CI.

We repeated the mediation analysis controlling for variables that have repeatedly been associated with alterations in brain structure. These included lifetime psychiatric diagnoses (mood [115,116,117,118,119], anxiety [120,121,122], and substance use disorder [123,124], Appendix A), warzone-related stress [125,126,127,128], body mass index (BMI) [129,130,131,132,133,134], current psychiatric medication use [135,136,137,138,139,140], race (white, non-white) [141], and completed years of education [142,143].

## 3. Results

For demographic information, please see Table 1 and Appendix A. The four groups did not significantly differ in age, the number of deployments, and the total duration of the deployment. Veterans with comorbid PTSD+mTBI were the most severely clinically burdened group, as indicated by the high number of comorbid psychiatric diagnoses, medication use, and the highest rates of military mTBI (mTBIs sustained during deployment or military service) in this group.

### 3.1. Group Differences in Sleep Quality

We first examined the influence of a diagnosis of PTSD and mTBI on sleep quality using ANCOVAs. Table 2 and Appendix A display the differences in sleep quality between the groups. The PTSD and comorbid PTSD+mTBI groups demonstrated more significant impairments on the PSQI global sleep quality, sleep efficiency, perceived sleep quality, and daily disturbances scales than those with mTBI or no history of PTSD or mTBI. There were no significant differences in sleep quality between the PTSD and comorbid PTSD+mTBI groups. Moreover, there was no significant difference in sleep quality between veterans with mTBI and veterans without a history of PTSD and mTBI.

### 3.2. PTSD, mTBI, and Sleep Quality

The regression analyses revealed a significant association between PTSD symptom severity and poorer global sleep quality in the total sample (*β* = 0.58, *t* = 9.15, *p* < 0.001), whereas there was no significant association between mTBI burden and global sleep quality (*β* = 0.07, *t* = 1.07, *p* = 0.288). Post-hoc analyses demonstrated that in the PTSD, mTBI, and comorbid PTSD+mTBI groups, more severe PTSD symptoms were associated with poorer global sleep quality. Moreover, in the PTSD and comorbid PTSD+mTBI groups, more severe PTSD symptoms were associated with poorer perceived sleep quality and more daily disturbances. In the mTBI group, more severe PTSD symptoms were associated with lower sleep efficiency (Appendix A).

### 3.3. Sleep Quality and WM Microstructure

The regression analyses investigating the association between sleep quality and WM microstructure showed a significant negative association between the PSQI global score and whole-brain FA_T_ in the total sample (*β* = −0.24, *t* = −3.35, *p* = 0.001). Post-hoc analyses revealed a significant association between global sleep quality and whole-brain FA_T_ in the comorbid PTSD+mTBI group (*β* = −0.39, *t* = −4.04, *f*^2^ = 0.18, *p* < 0.001). Supplementary analyses of the main WM fiber tracts similarly showed significant associations between global sleep quality and WM FA_T_ in the comorbid PTSD+mTBI group (Appendix A). No region-specific pattern was observed, supporting our hypothesis that poor sleep quality may lead to widespread WM alterations. We subsequently assessed the association between the three PSQI sub-scales (sleep efficiency, perceived sleep quality, and daily disturbances) and whole-brain FA_T_ in the comorbid PTSD+mTBI group. Only perceived sleep quality was significantly associated with whole-brain FA_T_ (*β* = −0.43, *t* = −3.86, *f*^2^ = 0.21, *p* < 0.001, Appendix A, Figure 2).

### 3.4. Sleep Quality Mediates the Association between PTSD Symptom Severity and WM Microstructure

Given the significant association between PTSD symptom severity and perceived sleep quality and between perceived sleep quality and WM microstructure in the comorbid PTSD+mTBI group, we performed additional mediation analyses to assess whether perceived sleep quality mediates the association between PTSD symptom severity and whole-brain FA_T_. When not including sleep in our model we observed an effect of PTSD symptom severity on whole-brain FA_T_ (*b* = −0.00, *SE* = 0.00, *t*(91) = −2.88, *p* = 0.005, Figure 3 path c). When including perceived sleep quality as a mediator, the relationships between PTSD symptom severity and perceived sleep quality (*b* = 0.02, *SE* = 0.00, *t*(91) = 5.55, *p* < 0.001, Figure 3 path a), perceived sleep quality and whole-brain FA_T_ (*b* = −0.00, *SE* = 0.00, *t*(90) = −3.26, *p* = 0.002, Figure 3 path b), and the model’s total effect (*F*(3, 90) = 6.81, *R*^2^ = 0.19, *p* < 0.001) were significant. However, the direct effect of PTSD symptom severity on whole-brain FA_T_ was not statistically significant (*b* = −0.00, *SE* = 0.00, *t*(90) = −0.97, *p* = 0.331, Figure 3 path c’). The findings indicate that the association between PTSD symptom severity and whole-brain FA_T_ is statistically mediated by perceived sleep quality. When the mediation model included psychiatric comorbidities (anxiety, depression, and substance use disorder), warzone-related stress, BMI, psychiatric medication use, race, and education as additional covariates, results did not change significantly.

## 4. Discussion

The current study investigated relationships between PTSD, mTBI, sleep quality, and WM microstructure in veterans. We observed impaired sleep quality in veterans with PTSD and comorbid PTSD+mTBI compared to those with mTBI only or those without a history of PTSD or mTBI. Additionally, global and perceived sleep quality measures were associated with characteristics of WM microstructure in veterans with comorbid PTSD+mTBI. Most importantly, our findings suggest that perceived sleep quality may explain the association between PTSD symptom severity and WM microstructure in veterans with comorbid PTSD+mTBI. Thus, our findings indicate that sleep plays a central role in how psychological trauma affects brain health.

### 4.1. Group Differences in Sleep Quality

In line with previous investigations [69,144], we found that individuals with PTSD or comorbid PTSD+mTBI experience poorer global sleep quality, sleep efficiency, perceived sleep quality, and more daily disturbances than individuals with mTBI only or no history of PTSD or mTBI. Contrary to earlier studies [11,145], we did not see a difference in sleep quality between veterans with mTBI and those without a history of PTSD or mTBI. Given that some of the participating veterans may have sustained their head trauma years ago, an explanation for this finding could be that sleep quality disturbances related to mTBI improved over time. Indeed, only a minority of individuals who sustain mTBI experience ongoing post-concussive symptoms (including sleep quality disturbances) [146,147,148]. On the contrary, recurrent sleep quality disturbances are still prevalent in individuals with remitted PTSD [17,18]. Therefore, PTSD appears to be an index of poor sleep quality, even without comorbid mTBI.

### 4.2. PTSD, mTBI, and Sleep Quality

We observed a significant association between greater PTSD symptom severity and worse global sleep quality, sleep efficiency, perceived sleep quality, and daily disturbances in veterans with PTSD, mTBI, and comorbid PTSD+mTBI. Interestingly, there was no significant association between mTBI burden and sleep quality, suggesting that the psychological consequences after a traumatic experience (rather than the physical trauma) are most predictive of sleep quality disturbances. These results highlight an integral role of poor sleep quality in PTSD severity [9,10], underscoring that traumatic experiences might be the driving force behind sleep quality disturbances in veterans [144,149].

### 4.3. Sleep Quality and WM Microstructure

As hypothesized [55,56,57,58,63], we showed a significant relationship between impaired sleep quality and characteristics of the WM microstructure. We speculate that the observed WM microstructure alterations might be due to impaired myelin repair processes. Previous research demonstrates that sleep initiates myelin deposition and repair [20] and is necessary for maintaining WM health. Myelin genesis and repair depend on the sufficient clearance of brain waste products [150]. The brain’s waste clearance system relies on the glymphatic system which consists of perivascular spaces vital for flushing out accumulated neurotoxins, such as beta-amyloid and tau [151,152]. Critically, the glymphatic system is most active during sleep [150]. Thus, we hypothesize that poor sleep quality, as observed in this study, may be related to impaired clearance of neurotoxins, which, in turn, may lead to neurodegenerative processes, including impaired myelination. This hypothesis is supported by the fact that amyloid and tau deposition have previously been linked to WM damage in veterans with comorbid PTSD+mTBI [153,154].

Notably, the association between impaired sleep quality and abnormal WM microstructure pertained solely to veterans with comorbid PTSD+mTBI. This finding may be ascribed to a statistical power effect as veterans with comorbid PTSD+mTBI made up the largest group. However, even though sleep quality disturbances were not statistically different between the comorbid PTSD+mTBI and PTSD groups, veterans with comorbid PTSD+mTBI presented as the most severely clinically burdened group, as indicated by the high number of comorbid psychiatric diagnoses, medication use, and the highest rates of military mTBI. Previous studies suggest that psychiatric disorders, medication use, and mTBI may increase brain vulnerability [51,71,155,156], potentially creating a neural environment that leaves the brain unprotected from the harmful effects of impaired sleep quality. Similarly, poor sleep quality negatively impacts brain structure and function, thus fueling the onset or progression of neuropsychiatric disorders and related brain abnormalities [13].

When assessing different aspects of sleep quality (sleep efficiency, perceived sleep quality, and daily disturbances), lower perceived sleep quality was the only significant indicator of alterations in WM microstructure. This finding aligns well with a previous study, reporting an association between overall sleep quality and cortical GM volume that was driven by perceived sleep quality [22]. Additionally, perceived sleep quality has been shown to be essential for functional outcomes and mental well-being in individuals with PTSD [16]. Of particular interest is that perceived sleep quality may not correlate with objectively measured sleep quality. Indeed, the phenomenon of paradoxical insomnia—the discrepancy between subjective and objective assessments of sleep [157]—is a common observation among veterans with sleep disorders [158] and PTSD [159]. Paradoxical insomnia is associated with general distress, ongoing hyperarousal states, and a negative cognitive bias that affects sleep perception [158,160,161,162,163,164]. Importantly, however, perceived sleep quality (rather than the objective assessment of sleep) appears to be reflective of overall mental well-being [164,165,166,167].

### 4.4. Sleep Quality Mediates the Association between PTSD Symptom Severity and WM Microstructure

We observed a relationship between more severe PTSD symptoms and greater WM abnormalities in the comorbid PTSD+mTBI group, which aligns well with several previous studies [36,53]. Strikingly, when including PTSD symptom severity, perceived sleep quality, and WM in the same statistical model, we found that perceived sleep quality accounted for the relationship between PTSD symptom severity and WM microstructure in veterans with comorbid PTSD+mTBI. This result suggests that poor sleep might be the most impactful symptom in the context of brain structure in individuals with comorbid PTSD+mTBI.

Future research is needed to elucidate the bidirectional interplay between sleep impairments and PTSD symptom severity. It is noteworthy that current first-line treatments for PTSD commonly fail to resolve sleep issues completely, even when other PTSD symptoms remit [18]. Persistent sleep quality disturbances are, in turn, a risk factor for PTSD [168], resulting in adverse bi-directionally reinforcing conditions [18]. Thus, interventions that reduce disturbed sleep quality may simultaneously improve overall PTSD symptom severity [18], given that restorative sleep is needed for fear extinction [169,170] and facilitates the emotional processing of traumatic events [171]. Targeting sleep disturbances is often a necessary first step when beginning trauma therapy to support emotional coping mechanisms and cognitive resources needed for a successful outcome [172]. Sleep-targeted interventions may yield higher acceptance and compliance, as they are less stigmatized than mental health therapies and may encourage more veterans to seek help if needed. Finally, specific training targeted at establishing and maintaining a healthy sleep routine even before deployment may be beneficial in fostering resiliency in veterans.

### 4.5. Limitations and Future Directions

We acknowledge several study limitations. First, our findings are limited to a male sample of veterans. They may, thus, not be generalizable to the general population of veterans, including women, given that sleep and WM structure may be affected by sex [173,174]. Second, the unequal sample sizes across groups may have affected statistical power and type I error rates, warranting replications utilizing balanced designs and larger samples. As highlighted above, the comorbid PTSD+mTBI group is the largest group, which might partially drive our findings. However, it is also critical to note that individuals with PTSD+mTBI were the most severely affected by psychiatric symptoms and that our results align with previous studies, suggesting that this group is specifically vulnerable. Third, mTBI diagnosis was based on retrospective self-recall of head injuries without available medical records for verification, potentially distorting reports of mTBI occurrence and severity. Similarly, sleep quality was assessed through self-report only and may not accurately reflect objective sleep quality. Future studies may benefit from employing trained clinicians to diagnose sleep disorders and include objective sleep measures, such as polysomnography, to record sleep quality complaints. Nevertheless, and as discussed above, subjective sleep quality serves as a valuable diagnostic tool indicative of mental and brain health. Furthermore, while we controlled for many potentially confounding variables, such as age, psychiatric comorbidities (anxiety, depression, and substance use disorder), warzone-related stress, BMI, psychiatric medication use, race, and education, we were unable to include other potentially relevant variables, such as caffeine or other stimulant use, exercise, cumulative sleep deficit, previous shift work, or socioeconomic status. Last, while the mediation analysis allowed for an advanced statistical assessment of complex interactions between the studied variables, the cross-sectional design restricts the interpretability of causal relationships. It is probable that some veterans experienced sleep quality disturbances even before deployment and were, thus, more likely to develop neuropsychiatric symptoms and exhibit structural brain alterations [168,175,176,177,178]. In summary, future longitudinal studies are needed to elucidate the underlying pathomechanism of perceived sleep quality and investigate its relationship with objective sleep quality and brain structure in veterans with PTSD and mTBI.

## 5. Conclusions

Findings from this study suggest that perceived sleep quality plays a vital role in mental and brain health in veterans. Importantly, our findings suggest that disturbed sleep quality may account for the relationship between PTSD symptom severity and WM microstructure alterations, which we speculate to result from impaired myelin repair processes, given that healthy sleep is required for lipid production and the proliferation of oligodendrocyte precursors, which are essential for myelin genesis and deposition. Furthermore, poor sleep has been linked to inadequate brain waste clearance through the perivascular glymphatic system, and accumulated neurotoxins may trigger neurodegenerative processes, including demyelination. Notably, our study reports a link between white matter alterations and perceived sleep quality. While self-reported sleep quality is a strong indicator of mental well-being, it may not necessarily mirror objectively assessed sleep quality. Future research may benefit from employing both self-reports and device-assessed sleep ratings to gain further insights into sleep disturbances in relation to PTSD and mTBI. Moreover, future research is needed to investigate whether sleep-targeted interventions may benefit overall brain health in the veteran population.

## Figures and Tables

**Figure 1 jcm-12-02079-f001:**
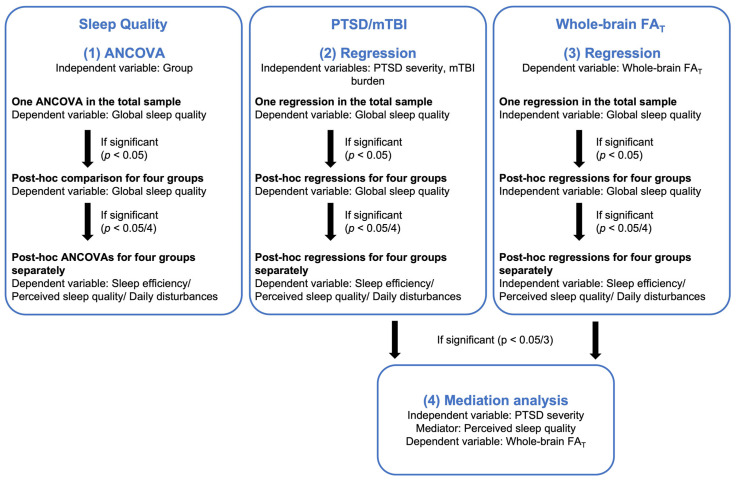
Hierarchical statistical approach. Note. PTSD, post-traumatic stress disorder; mTBI, mild traumatic brain injury; PSQI, Pittsburgh Sleep Quality Index; FA_T_, fractional anisotropy tissue. This figure illustrates the hierarchical statistical approach.

**Figure 2 jcm-12-02079-f002:**
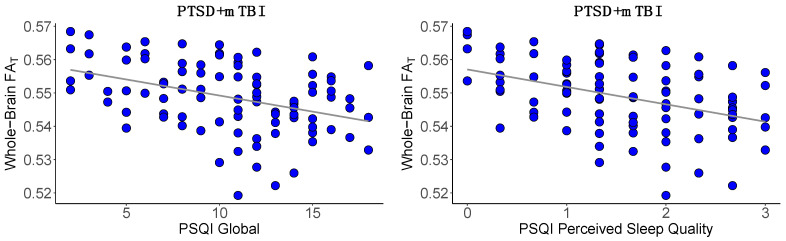
Association between sleep quality and whole-brain FA_T_. Note. PSQI, Pittsburgh Sleep Quality Index [76]; PTSD, post-traumatic stress disorder; mTBI, mild traumatic brain injury; FA_T_, fractional anisotropy tissue. This figure illustrates the significant negative association between global sleep quality and whole-brain FA_T_ (*p* < 0.001) and between perceived sleep quality and whole-brain FA_T_ (*p* < 0.001). Lower scores on the PSQI scales represent better sleep quality.

**Figure 3 jcm-12-02079-f003:**
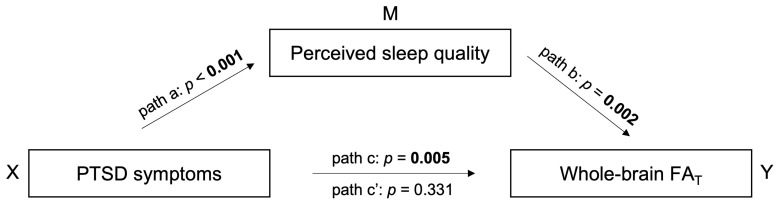
Mediation model. Note. PTSD, post-traumatic stress disorder; mTBI, mild traumatic brain injury; FA_T_, fractional anisotropy tissue. This figure illustrates the mediating effect of perceived sleep quality between PTSD symptom severity and whole-brain FA_T_. Path a refers to the association between X and M. Path b refers to the association between M and Y when taking X into account. Path c represents the total effect of X on Y, including the axb path. Path c’ shows the direct effect of X on Y when M is omitted.

**Table 1 jcm-12-02079-t001:** Sample characteristics.

		Total Sample (*N* = 180)		PTSD (*n* = 38)		mTBI (*n* = 25)		Comorbid PTSD+mTBI (*n* = 94)		No History of PTSD or mTBI *(n* = 23)	
		Mean ± SD	Range	Mean ± SD	Range	Mean ± SD	Range	Mean ± SD	Range	Mean ± SD	Range
**Age (years)**		31.49 ± 7.60	20–55	31.18 ± 6.79	22–48	32.60 ± 9.82	21–55	31.22 ± 7.18	20–53	31.87 ± 8.24	23–53
**Education (years)**		13.89 ± 1.96	12–20	13.34 ± 1.60	12–17	14.36 ± 1.78	12–18	13.84 ± 2.03	12–20	14.48 ± 2.19	12–19
**Number of OEF/OIF deployments**		1.40 ± 0.69	1–5	1.39 ± 0.79	1–4	1.40 ± 0.65	1–3	1.41 ± 0.71	1–5	1.35 ± 4.87	1–2
**Number of other stressful deployments**		0.39 ± 0.77	0–5	0.37 ± 0.63	0–3	0.24 ± 0.44	0–3	0.49 ± 0.89	0–5	0.22 ± 0.52	0–2
**Total duration of OEF/OIF deployments (months)**		13.64 ± 8.37	3–53	13.84 ± 9.75	5–53	14.72 ± 7.89	3–31	13.49 ± 8.36	3–52	13.87 ± 6.81	4–28
**Total duration of other deployments (months)**		2.84 ± 6.21	0–49	3.03 ± 5.22	0–17	1.36 ± 3.87	0–16	3.41 ± 7.24	0–49	1.83 ± 4.93	0–18
**Number of lifetime mTBIs**		1.39 ± 2.18	0–18	0.00 ± 0.00	0–0	1.56 ± 0.96	1–5	2.24 ± 2.62	1–18	0.00 ± 0.00	0–0
**Lifetime mTBI burden**		2.22 ± 3.42	0–31	0.00 ± 0.00	0–0	2.40 ± 1.47	1–6	3.61 ± 4.10	1–31	0.00 ± 0.00	0–0
**Total CAPS**		45.84 ± 27.14	0–104	39.79 ± 21.15	6–88	15.84 ± 13.73	0–41	51.80 ± 18.65	2–92	12.65 ± 10.60	0–39
		** *n* **	**% ***	** *n* **	**% ***	** *n* **	**% ***	** *n* **	**% ***	** *n* **	**% ***
**Ethnicity**	American Indian or Alaska Native	1	0.56	0	0.00	0	0.00	1	1.06	0	0.00
	Asian	2	1.11	1	2.63	0	0.00	1	1.06	0	0.00
	Black	13	7.22	4	10.53	3	12.00	6	6.38	0	0.00
	Hispanic or Latino	26	14.44	6	15.79	5	20.00	13	13.83	2	8.70
	Native Hawaiian or Pacific Islander	0	0.00	0	0.00	0	0.00	0	0.00	0	0.00
	White	138	76.67	26	68.42	17	68.00	74	78.72	21	91.30
	Unknown	1	0.56	1	2.63	0	0.00	0	0.00	0	0.00
**Service branch**	Army	33	18.33	4	10.53	6	24.00	21	22.34	2	8.70
	Army National Guard	62	34.44	15	39.47	10	40.00	26	27.66	11	47.83
	Air Force	9	5.00	1	2.63	2	8.00	5	5.32	1	4.35
	Air Force National Guard	7	3.89	2	5.26	1	4.00	3	3.19	1	4.35
	Coast Guard	0	0.00	0	0.00	0	0.00	0	0.00	0	0.00
	Navy	7	3.89	1	2.63	1	4.00	5	5.32	0	0.00
	Marines	38	21.11	5	13.16	4	16.00	27	28.72	2	8.70
	Reserves	25	13.89	10	26.32	2	8.00	9	9.57	4	17.39
	National Guard, branch unknown	6	3.33	1	2.63	0	0.00	3	3.19	2	8.70
**Psychiatric diagnoses**	Lifetime PTSD	132	73.33	38	100.00	0	0.00	94	100.00	0	0.00
	Lifetime substance use disorder	118	65.56	23	60.53	13	52.00	72	76.60	10	43.48
	Lifetime mood disorder	63	35.00	14	36.84	4	16.00	42	44.68	3	13.04
	Lifetime anxiety disorder	30	16.67	9	23.68	3	12.00	15	15.96	3	13.04
**Psychiatric medication**	Antidepressants	37	20.55	8	21.05	1	4.00	28	29.79	0	0.00
	Antiseizure medication	12	6.67	2	5.26	0	0.00	10	10.64	0	0.00
	Sedatives	13	7.22	5	13.16	1	4.00	7	7.45	0	0.00
	Pain medication	61	33.89	10	26.32	10	40.00	38	40.43	3	13.04
	Prazosin	5	2.78	0	0.00	0	0.00	5	5.32	0	0.00
**mTBI**	Military mTBI	64	35.56	0	0.00	7	28.00	57	60.64	0	0.00
	Lifetime mTBI	119	66.11	0	0.00	25	100.00	94	100.00	0	0.00

Note. SD, Standard deviation; PTSD, post-traumatic stress disorder; mTBI, mild traumatic brain injury; OEF, Operation Enduring Freedom; OIF, Operation Iraqi Freedom; CAPS, Clinician Administered PTSD Scale [73]; PSQI, Pittsburgh Sleep Quality Index [76]; PSQI 3-factor structure, PSQI subscales sleep efficiency, perceived sleep quality & daily disturbances [77]. % ***** Percentage of total cases per group.

**Table 2 jcm-12-02079-t002:** Sleep quality group comparisons.

		Total Sample			PTSD			mTBI			Comorbid PTSD+mTBI			No History of PTSD or mTBI		
		*N*	Mean ± SD	Range	*n*	Range	Mean ± SD	*n*	Range	Mean ± SD	*n*	Range	Mean ± SD	*n*	Range	Mean ± SD
**PSQI Global**		178	8.97 ± 4.46	0–20	38	8.71 ± 4.43	0–20	25	6.68 ± 3.61	1–18	92	10.49 ± 4.23	2–18	23	5.78 ± 3.48	0–14
**PSQI 3-factor structure**	Sleep efficiency	179	1.23 ± 0.99	0–3	38	1.13 ± 0.98	0–3	25	0.86 ± 0.99	0–3	93	1.46 ± 0.94	0–3	23	0.89 ± 1.07	0–3
	Perceived sleep quality	180	1.31 ± 0.80	0–3	38	1.30 ± 0.73	0–3	25	0.99 ± 0.64	0–2.67	94	1.54 ± 0.82	0–3	23	0.72 ± 0.52	0–2
	Daily disturbances	179	1.32 ± 0.62	0–3	38	1.28 ± 0.59	0–2.5	25	1.00 ± 0.48	0–2	93	1.53 ± 0.61	0–3	23	0.91 ± 0.47	0–2
		**PTSD vs. mTBI vs.** **Comorbid PTSD+mTBI vs.** **No history of PTSD or mTBI**			**Post-hoc** **PTSD vs.** **mTBI**		**Post-hoc** **PTSD vs. Comorbid PTSD+mTBI**		**Post-hoc** **mTBI vs. Comorbid PTSD+mTBI**		**Post-hoc** **PTSD vs.** **No history of PTSD or mTBI**		**Post-hoc** **mTBI vs.** **No history of PTSD or mTBI**		**Post-hoc** **Comorbid PTSD+mTBI vs. No history of PTSD or mTBI**	
		**ANCOVA**														
		***F*(df),**	** *p* **	** *η* ^2^ **	***p* ***											
**PSQI Global**		11.430(3, 173)	**<0.001**	0.17	0.055		0.026		**<0.001**		**0.008**		0.457		**<0.001**	
**PSQI 3-factor structure**	Sleep efficiency	4.16(3, 174)	**0.007**	0.07	0.259		0.078		**0.006**		0.339		0.894		**0.012**	
	Perceived sleep quality	9.20(3, 175)	**<0.001**	0.14	0.112		0.091		**0.001**		**0.004**		0.224		**<0.001**	
	Daily disturbances	10.66(3, 174)	**<0.001**	0.16	0.057		0.025		**<0.001**		0.017		0.618		**<0.001**	

Note. SD, Standard deviation; PTSD, post-traumatic stress disorder; mTBI, mild traumatic brain injury; PSQI, Pittsburgh Sleep Quality Index [76]; PSQI 3-factor structure, PSQI subscales sleep efficiency, perceived sleep quality & daily disturbances [77]. All ANCOVA’s were corrected for age. *p* * corrected for multiple comparisons as outlined in Figure 1.

## Data Availability

The data presented in this study is available on request from the corresponding author.

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
