# Peer review of "Sleep Quality Disturbances Are Associated with White Matter Alterations in Veterans with Post-Traumatic Stress Disorder and Mild Traumatic Brain Injury"

_jcm, 2023, doi:10.3390/jcm12052079_

Round 1
Reviewer 1 Report
Thank you very much for the chance of reviewing this meaningful manuscript which evaluated the association between sleep quality and PTSD symptoms among veterans. Overall, the quality is high. Only several issues may need to be addressed.
In the introduction, the contributions of this study can be stated. The significance of this study can be emphasized.
How do you decide the sample size? Based on what rationales?
The conclusion is a bit short. I believed more can be added to enrich the content.
Reviewer 2 Report
Dear Authors,
Congratulations for excellent work. I found Your study properly performed and with scientifically sound results. I have only some minor questions/remarks that in my opinion may help in underlining/clarifying the results:
1. Did You collect data on symptoms of disordered sleep or somnological diagnosis (e.g. insomnia, daytime sleepiness, parasomnias). If so, I believe that adding to that study data on prevalence of specific sleep disorders would underline significance of Your study.
2. THere is one unclear description (line 333): "In the mTBI group, more severe PTSD symptoms were associated with lower sleep efficiency" . As I understood it was pure mTBI subgroup with no diagnosis of PTSD so why those patients had PTSD symptoms?
3. You have suggested (Discussion) that mTBI-related sleep disorders might have improved over time. Are data on time passed from the last mTBI available? IF so - maybe it would be useful including that factor into analysis.
4. The mTBI+PTSD subgroup suffered from most severely worsened group. Meanwhile this group was also the one most affected with burden of psychiatric disorders. Was that burden included into statistical models depicting sleep quality-PTSD-FA relations?
